# Lynch Syndrome Biopathology and Treatment: The Potential Role of microRNAs in Clinical Practice

**DOI:** 10.3390/cancers15153930

**Published:** 2023-08-02

**Authors:** Serena Ascrizzi, Grazia Maria Arillotta, Katia Grillone, Giulio Caridà, Stefania Signorelli, Asad Ali, Caterina Romeo, Pierfrancesco Tassone, Pierosandro Tagliaferri

**Affiliations:** 1Department of Experimental and Clinical Medicine, Magna Graecia University of Catanzaro, 88100 Catanzaro, Italy; serena.ascrizzi@studenti.unicz.it (S.A.); arillottagraziamaria@gmail.com (G.M.A.); k.grillone@unicz.it (K.G.); g.carida@unicz.it (G.C.); stefania.signorelli@unicz.it (S.S.); asad.ali@studenti.unicz.it (A.A.); cateromeo93@hotmail.it (C.R.); tassone@unicz.it (P.T.); 2Medical Oncology and Translational Medical Oncology Units, University Hospital Renato Dulbecco, 88100 Catanzaro, Italy

**Keywords:** Lynch syndrome, hereditary non-polyposis colorectal cancer, HNPCC, microRNAs, miRNAs, non-coding RNAs, DNA-damage repair, MSI, hereditary cancer, colorectal cancer

## Abstract

**Simple Summary:**

Lynch syndrome is an autosomal dominant hereditary disease that confers a high risk of developing various types of tumors, especially colorectal and endometrial cancer. In recent years, the molecular mechanisms underlying Lynch syndrome have generated considerable interest. Several studies have highlighted the epiphenomenon of microsatellite instability, caused by a deficit in the ability of cells to repair DNA damage, as a disease-specific feature. The discovery of microRNAs has served to clarify how these small oligonucleotide molecules may contribute to the progression of Lynch syndrome by modulating the expression of several players involved in DNA repair pathways. The purpose of this review is to analyze the management of patients with Lynch syndrome by emphasizing the importance of microRNAs as markers or therapeutics for the development of novel clinical approaches.

**Abstract:**

Lynch syndrome (LS), also known as Hereditary Non-Polyposis Colorectal Cancer (HNPCC), is an autosomal dominant cancer syndrome which causes about 2–3% of cases of colorectal carcinoma. The development of LS is due to the genetic and epigenetic inactivation of genes involved in the DNA mismatch repair (MMR) system, causing an epiphenomenon known as microsatellite instability (MSI). Despite the fact that the genetics of the vast majority of MSI-positive (MSI^+^) cancers can be explained, the etiology of this specific subset is still poorly understood. As a possible new mechanism, it has been recently demonstrated that the overexpression of certain microRNAs (miRNAs, miRs), such as miR-155, miR-21, miR-137, can induce MSI or modulate the expression of the genes involved in LS pathogenesis. MiRNAs are small RNA molecules that regulate gene expression at the post-transcriptional level by playing a critical role in the modulation of key oncogenic pathways. Increasing evidence of the link between MSI and miRNAs in LS prompted a deeper investigation into the miRNome involved in these diseases. In this regard, in this study, we discuss the emerging role of miRNAs as crucial players in the onset and progression of LS as well as their potential use as disease biomarkers and therapeutic targets in the current view of precision medicine.

## 1. Introduction

Lynch syndrome (LS), also named hereditary non-polyposis colorectal cancer (HNPCC), is an autosomal dominant disease that can lead to a high risk of various cancer types. The most frequent malignancy developed by LS patients is colorectal cancer (CRC) (LS I) [1,2,3], which occurs in about 50–80% of cases [1,4]. Moreover, LS patients present a high probability of developing extracolonic malignancies [5], mainly endometrial cancer (40–60% of cases) [4] (LS II) [6] but also cancers of the small bowel [7], genitourinary tract (ureter, renal pelvis, bladder, prostate), stomach, hepatobiliary tract, pancreas, ovary, breast [8,9,10], and brain (glioblastoma, Turcot’s syndrome) [11,12,13]. In addition to malignancies, LS patients may also present benign lesions, such as sebaceous [2] and keratoacanthoma (Muir-Torre syndrome) [14].

The main molecular event underlying LS carcinogenesis is the mismatch repair deficiency (dMMR), which causes the epiphenomenon of microsatellite instability (MSI). MSI is a condition characterized by errors in the replication and repair process that causes alterations in the length of microsatellites, short repetitive DNA sequences that remain stable in MMR-proficient cells [15]. At present, the assessment of dMMR is recognized to have clinical, diagnostic, therapeutic, and surveillance implications in patients with LS [16]. LS frequency is about 1 out of 440 and varies among different populations [16,17]. More than 50 mutations in MMR genes have recently been identified in Icelandic, French Canadian, African American, Polish, and Latin American populations [18,19,20]. However, the estimated frequency of LS in the population is limited by the availability of data only on patients with a history of cancer [16].

The diffusion of highly sophisticated technologies to dissect the molecular landscape of malignancies through multi-omic approaches [21,22], as well as the growing evidence regarding the crucial role of non-coding RNAs (ncRNA) in cancer development and treatment responses [23,24], has prompted a deeper investigation into the contribution of ncRNAome to several diseases, including LS. NcRNAs are classified into micro or long ncRNAs based on their length (<200 nucleotides or >200 nucleotides, respectively). Here we focused on the involvement of microRNAs (miRNAs, miRs) in LS onset and progression.

MiRNAs are one of the most abundant endogenous regulatory molecules that can influence various biological processes [25,26]. Studies on miRNAs have revealed how they are involved in the pathogenesis of many human diseases, including cardiovascular and neurological disorders, with special relevance in cancer development, including hematopoietic and solid malignancies [27]. By investigating miRNA expression in cancer cells compared to that of normal cells, significant differences were observed [28]. Based on their expression profiling and functional roles in tumor cells, miRNAs can be classified as oncogenes (onco-miRs) or tumor suppressors (tumor-suppressor miRs) [29]. Regarding MSI, research studies have demonstrated that some miRNAs can be differentially expressed in MSI-positive (MSI^+^) and MSI-negative (MSI^−^) tumors compared with microsatellite-stable (MSS) tumors [30,31]. Their ability to act as promoters or suppressors of cancer development by affecting the expression of multiple genes make them potential therapeutic targets alone or in combination with other therapeutics [32].

## 2. Lynch Syndrome

### 2.1. Genetic and Molecular Background

The main molecular events underlying LS are defects in the MMR system, which is a mechanism that prevents the accumulation of mutations due to DNA mismatch that may occur during cell division, including single nucleotide mismatches, insertions, and deletion loops [33,34]. The MMR system functions as a multienzyme complex capable of forming heterodimers to repair DNA damage and promote genomic stability [35,36] with high fidelity and efficiency [37].

MMR multi-enzyme complexes mainly include MSH2 and MLH1 proteins. These two proteins form protein heterodimers with other players. MSH2 associates with MSH6 or MSH3 (MutS α or MutS β); MLH1 associates with PMS2, PMS1, MLH3 (MutL α, MutL β, MutL γ) [38].

The process of damage repair can include three steps. The first step begins with the identification of DNA mismatch by the MutS and MutL complexes. Once the complexes are formed and the damaged region is recognized, the initiation complex is assembled, which consists of MutS α or β, MutL α (MLH1-PMS2), and PCNA (Figure 1A) [39]. EXO-1, DNA polymerase δ (Figure 1B), and DNA ligase are recruited to this complex, which make the cut in the damage sequence, correctly resynthesize the strand, and reconstitute the base bonds, respectively (Figure 1C) [40].

As previously observed, when the MMR system is defective, there is an accumulation of mutations, which causes the microsatellite instability epiphenomenon (MSI-H) and prompts the onset of several cancers, including LS [35,38].

Mutations in MLH1 and MSH2 are estimated to cause about 70% of LS cases [41]; MSH6 mutations cause up to 14%, and PMS2 mutations contribute to less than 15% [42]. The germinal deletion of EPCAM, a gene not directly belonging to MMR, inactivates MSH2, causing LS in 1–3% of cases [43,44]. However, the number of reported mutations affecting MMR genes varies considerably [45] because LS cases are diagnosed in the background of cancers whose variants may be different in terms of number, type, and the genes involved [46]. Mutations in MLH1 and MSH2 [47] are associated with a high risk of developing LS; mutations in MSH6 are associated with intermediate risk, and mutations in PMS2 are associated with low risk [48]. In the case of EPCAM, the presence of mutations are associated with a high risk of CRC development but low risk of extra-colic cancer development [49,50]. In any case, the overall risk of developing cancer in LS patients also depends on gender, age, history of malignancy, environmental factors, and lifestyle [45].

The malignant transformation in LS-related CRC has been described as a multistep process and explained by different models of progression (Figure 2). The first proposed model supposes that tumors arise from polypoid lesions with proficient MMR, whose deficiency will occur at later steps [51]. In particular, it was assumed that the formation of sporadic polyps with dysfunctional APC-mediated systems was the event promoting the adenoma to CRC formation and that the loss of MMR, promoting the accumulation of somatic mutations, was a subsequent event underlying the genesis of the invasive form (Figure 2A) [52,53]. Advances in molecular analysis technologies have led to the definition of a new model that completely overcomes adenomatous precursors to explain LS-associated CRC [54]. This new model is based on the detection of intestinal crypt foci with dMMR [55]. These foci are histologically normal and apparently non-neoplastic, which is common in the intestinal epithelium of healthy LS carriers without cancer. These crypts are adjacent to others with a functional dMMR, suggesting a role of this system in the adenoma initiation. These areas may acquire somatic mutations in TP53 or CTNNB1 by causing immediate and direct invasive tumor growth (Figure 2B) [56]. This model is able to explain the accelerated development of CRCs within a short interval between screening colonoscopies [57].

### 2.2. Clinical and Diagnostic Overview

From a pathological point of view, LS-related CRC is characterized by low differentiation (grade 4), mucinous or castellated ring cells, and a bone marrow growth pattern with rich Crohn-type lymphocyte infiltrates, representing a response to neoantigens generated by the high mutational burden [14,51,58].

From a genetic perspective [59], the phenotypic histopathological and clinical manifestations of LS mainly depend on the mutated gene responsible for MMR. For example, individuals with MSH2 mutations frequently develop genito–urinary tract and endometrial neoplasms [60], while individuals with MSH6 mutation are more prone to develop Breast Cancer (BC) [61]. Therefore, surveillance and treatment programs for these patients should be modulated based on the mutated genes [16].

CRC in patients with LS has different features than sporadic CRC [62]. Usually LS-related CRC localizes to the right/proximal side of the colon, manifest early (40–50 years old), and have propensity for synchronous and metachronous colorectal carcinoma, again according to genotypes [63]. For example, LS patients with PSM2 mutations [64] develop malignancies later than other patients, while patients with MSH6 mutations have CRC and endometrial carcinoma later in life compared with MLHI or MSH2 mutation carriers [47].

To date, early diagnosis in individuals with typical LS genetic mutation has a triple significance as (i) it allows for the development of a program of close surveillance of the patient themselves in view of their high risk of developing cancer; (ii) it is necessary to offer their family members appropriate screening checks; and, above all, (iii) it has proved to be essential because of the known therapeutic implications (avoid chemotherapy treatment with adjuvant purposes and the use of immune checkpoint inhibitors) [12,65].

In the past, both clinical and histopathological criteria have been used to diagnose LS [5]. The original clinical criteria were the Amsterdam I criteria, according to which the clinical diagnosis of LS could be made if the following conditions were met: the presence of three family members diagnosed with CRC in two successive generations (with one first-degree relative from the other two), at least one case of a family member with CRC before the age of 50, and the exclusion of familial adenomatous polyposis [66]. These criteria were modified in 1999 to include neoplasms of the extra-colonic district in the Amsterdam II criteria [67].

In addition to these purely clinical criteria, the Bethesda criteria were devised [68], which combine family history with histopathologic features typical of LS-related malignancies (mucinous histology, castellated ring cells, presence of abundant lymphocytic infiltrates, Crohn-like reactions, bone marrow growth pattern): CRC diagnosis at under 50 years, the presence of synchronous or metachronous colon cancers, diagnosis of colon cancer at 60 years of age but histologic features similar to MSI-H, at least one first-degree relative with Lynch-related cancer and/or CRC diagnosed when the relative is under 50 years of age, and two or more first- or second-degree relatives with Lynch-related cancers [69].

New risk prediction algorithms [5] such as PREMM [70,71,72], MMRpro [72], and MMRpredict [73] have recently been devised for LS. These quantify, in percentage, the risk of a patient carrying a mutation in the LS genes, therefore determining the need for genetic evaluation and possible germline genetic testing based on the patient’s personal and family history of cancer [16].

Notably, PREMM5 is the first clinical model based on the patient’s sex, age, and personal/family history to provide a risk assessment for all five LS genes (MSH2, EPCAM, MLH1, MSH6, and PMS2); a score of 2.5% or higher indicates a need for genetic evaluation [74].

However, all these clinical patterns of diagnosis have a very low detection rate. Less than 50% of LS patients meet these criteria [1]. This is due to limited knowledge and difficulty in reconstructing family history [14] but also due to wanting to identify higher-risk individuals with these criteria rather than the entire spectrum of LS patients [75].

Bethesda criteria combining clinical features with histopathologic features also demonstrated high sensitivity but low specificity [5].

Today, the diagnosis of LS is performed through an early evaluation of tumor tissue or by immunohistochemical analysis to demonstrate dMMR but also by using a polymerase chain reaction (PCR)-based test to demonstrate MSI status [14].

The results of these tests have been shown to be highly concordant [14], allowing for the identification of patients to be tested to investigate the germline variants of the selected genes involved in MMR [12].

The PCR test for MSI evaluation compares changes in short repeated sequences at different loci (usually at the five most frequent loci) [65]. The tumor tissue sample is considered MSI-High (MSI-H) if MSI is detected in at least two or more out of five loci (30%), MSI-Low (MSI-L) if MSI is detected in less than one out of five loci (less than 30%), and microsatellite stability (MSS) is deemed to have occurred if MSI is not detected in any of the loci [65,76]. MSI-H indicates a dMMR, while MSI-L or no MSI indicates efficient MMR repair [14]. Clinically, MSI-L and MSS behave similarly and are often clustered together [5].

However, it should be noted that most cancers with MSI-H are attributable to the somatic and/or epigenetic inactivation of genes involved in MMR rather than germline mutations [65]. Only 15% of CRC cases with MSI-H are due to LS [77]. The most frequent somatic mutation of MMR genes that results in MSI-H with CRC and endometrial development is promoter hypermethylation, which results in MLH1 silencing [65,78]. Since MLH1 and PMS2 proteins function as a stable heterodimer, the simultaneous loss of expression of both means that there is an underlying alteration of MLH1 caused by the somatic methylation of the MLH1 promoter (sporadic case) or by the germline mutation of MLH1 (LS) [14].

Considering the high-frequency somatic hypermethylation of MLH1 in CRC and endometrial cases, before germline testing and genetic evaluation, hypermethylation of the MLH1 promoter should be excluded either by direct evaluation or by assessing the presence of the somatic BRAF V600E mutation, which, as a result, acquires negative predictive values in the case of CRC [65,79].

However, compared with MLH1, the loss of expression of MSH2 or MSH6 (or both) and PMS2 is usually caused by a germline mutation. Therefore, patients with CRC and a loss of MLH1 with the absence of the BRAF V600E mutation or MLH1 hypermethylation, as well as patients with a loss of MSH2 or MSH6 (or both) or PMS2 and elevated MSI, should be referred for genetic counseling and germline genetic testing to confirm LS [14].

In the past few years, patients with colorectal cancer or MSI-H/dMMR endometrial carcinoma without germline mutations and MLH1 promoter hypermethylation were erroneously referred to as “presumed LS”. Nowadays, due to the development and increasing use of next-generation sequencing (NGS) on tumor tissue in clinical–diagnostic practice to determine MMR status [80], they are instead referred to as “Lynch-like syndrome” [14]. Evidence of dMMR/MSI-H due to the biallelic somatic mutations of MMR genes or single somatic mutations with heterozygous loss of the other allele have been reported [81].

The identification of Lynch-like syndrome has a very important therapeutic implication since these patients can benefit from treatment with immune checkpoint inhibitors [14].

NGS detects tumor mutational burden (TMB), also identifying therapeutic targets, as in the case of colorectal mutations affecting genes involved in the RAS pathway [14]. When NGS detects a suspected germline mutation, confirmatory sequencing of the germline should be performed via orthogonal sequencing [14]. Today, NGS is increasingly used because its multigene panels offer numerous advantages, such as the following: identifying individuals with LS and atypical clinical phenotypes, identifying mutations in genes associated with the risk of developing cancer, identifying target mutations but also having lower cost content, and speed in the availability of results [82].

Recent data also show that somatic NGS panels have concordant results with traditional immunohistochemical and molecular PCR assays for the evaluation of MSI/dMMR [83]. These NGS panels also seem to overcome some of the biases of the PCR assay, and it has been suggested that they may replace them in the future, especially when implemented for the detection of other relevant somatic mutations [82].

Although it is not yet widespread clinical practice, regardless of the test used, a screening for LS should be performed whenever a new cancer diagnosis is made for therapeutic implications, for the prevention of further cancer in the same patient, and for the prevention of primary cancers in the patient’s family members [65].

Today, guidelines recommend (i) screening all diagnosed colorectal and endometrial cancer patients for LS; (ii) excluding hypermethylation of the MLH1 promoter, MMR-D/MSI-H, CRC, and/or BRAF V6OOE mutation in the case of MLH1 and PMS2 expression; and (iii) offering patients genetic evaluations and possible confirmatory germline testing.

### 2.3. LS and Breast Cancer: What Is the Real Link?

Breast cancer (BC) is one of the cancers associated with LS, as reported by some studies suggesting that women with LS may have a two- to three-fold increased risk of developing BC than the general population.

To date, the relationship between LS and BC remains much debated. The available data lack statistical significance because of the small sample sizes, the type of retrospective studies, and the fact that only epidemiological analyses are available [84]. However, the onset of BC in LS patients is around 50 years of age; therefore, mammography is still strongly recommended as a screening technique for the general population [85].

In patients with LS with BC, genetic testing and counseling are critical for estimating the risk of developing other cancers and choosing appropriate screening and management strategies. It would be interesting to more thoroughly investigate the link between BC and LS in order to correlate them with a unique BC histotype and/or phenotype (Her2+? Triple negative?). It should be mentioned that a recent molecular approach has demonstrated an increased risk of developing BC in patients with LS secondary to MSH6 and PMS2 germline mutations [61].

Moreover, the potential relationship between LS-BC and MSI could confirm an already established therapeutic strategy [86]. As a matter of fact, MSI is recognized as a predictive biomarker of response to immunotherapy [87]. Therefore, immunohistochemical testing for MSI on tumor tissue could also be considered in the case of BC.

While regular screening and genetic counseling can help manage risk and prevention strategies for people with LS and BC, the characterization of these patients’ MMR status is difficult due to the absence of tumor-specific guidelines and/or companion diagnostic tests. Therefore, accurately identifying dMMR-BC can be a challenge [88].

## 3. miRNAs

### 3.1. Biogenesis and Mechanism of Action

MiRNA-encoding sequences are in the intronic regions of coding genes, but some of them can be synthesized from exonic regions by RNA polymerase II (RNA pol II) [89]. The miRNA coding sequences are close to each other, thus establishing a single polycistronic transcription unit [90].

Chromatin studies have been carried out to identify promoter regions to understand the mechanism of miRNA transcriptional regulation. It was found that some miRNAs have their own promoter sites, while others appear to be regulated by the promoter region of coding genes [91].

As in the case of mRNAs, miRNAs also proceed through different levels of maturation. Pri-miRNAs are transcribed by RNA pol II and then processed by a microprocessor complex consisting of a DiGeorge syndrome RNA-binding protein (DGCR8) and a ribonuclease III enzyme (Drosha) [92]. DGCR8 recognizes N6-methyladenylated GGAC motifs, while Drosha eliminates secondary structures that may be formed along the pri-miRNA [93,94], generating the pre-miRNA. Through the exportin 5 (XPO5)/RanGTP transport system, the pre-miRNA translocate from the nucleus to the cytoplasm [95]. In the cytoplasm, RNase III Dicer endonuclease recognizes and cuts the hairpin structures of the pre-miRNA, generating a duplex strand that is about 22 nucleotides long with protruding ends at 2′ and 3′ [96]. Mature miRNAs are loaded by Dicer and other proteins into Argonaute (AGO) proteins [92].

The complex formed by mature miRNAs and AGO proteins is known as an RNA-induced silencing complex (RISC) and is implicated in gene silencing at both transcriptional and post-transcriptional levels by binding the target messenger RNA (mRNA) [97,98].

In recent decades, miRNAs have been the focus of several studies that allowed for the identification of the molecular mechanisms that mediate gene silencing, mainly based on the recognition of the target mRNA via Watson and Crick complementarity.

Gene silencing begins with the generation of the minimal mRNA silencing complex (miRISC) formed by the mature miRNA and AGO proteins, which function as the guide strands [99,100]. The AGO proteins also represent the catalytic core of the miRISC complex. In fact, some AGO protein families have been shown to be capable of slicing and, consequently, initiating mRNA degradation. At the basis of this activity is the high complementarity between the guide strands and the target mRNA sequence [101].

The sequence that is complementary to the guide strands, present on the mRNA, is called miRNA response elements (MREs) [99], and it is often located in the 3′UTR. Interaction between miRNA and MREs located in the 3′UTR induces deadenylation and the decapping of the mRNA [99]. Furthermore, algorithm-based computational studies have shown that miRNAs can induce mRNA silencing by also binding other regions [102]. For example, it has been investigated how miR-532-5p has an oncogenic function in in vitro models of CRC by binding to the 5′UTR of RUNX3 (tumor suppressor) mRNA [103]. Another class of miRNAs acts by binding both UTRs of the mRNA. This is the case for hsa-miR-34, which simultaneously binds the 3′ UTR and 5′ UTR of AXIN2 mRNA, providing stronger silencing [104]. MiRNAs may localize in the cytoplasm in various organelles, and once matured in the cytoplasm, they may also be bound by Importin-8 and transported to the nucleus [105]. There, the nuclear miRISC complex can interact with target mRNAs in a co/post-transcriptional manner, also acting on the splicing or alternative splicing mechanism [106,107]. Some miRNAs can also perform a transcription factor-like function by intervening in the chromatin remodeling process, binding to the promoter sites of certain genes, or interacting at specific loci that transcribe enhancer-derived RNAs (eRNAs) that intensify mRNA transcription [108].

### 3.2. Colorectal Cancer and miRNAs

CRC is the third most common cancer worldwide and has a high mortality rate [109]. Several studies have aimed to identify the risk factors (genetic and environmental) that contribute to the onset and progression of this type of cancer [110]. Due to its high frequency in the population and its high mortality rate, major efforts are being devoted to identifying new therapeutic targets and prognostic biomarkers of the disease.

The biological relevance and clinical implications of miRNA families have been increasingly explored in recent years to shed light on genomic “dark matter”, including miRNAs [22].

In accordance with the literature, several classes of miRNAs were identified as playing a crucial role in the onset and progression of CRC and as being deregulated in comparison to normal tissues [111].

The abnormal expression of specific miRNAs in CRC was associated with the deregulation of key biological processes, including proliferation, metastasis, angiogenesis, autophagy, apoptosis, and chemoradiotherapy resistance [111].

The monitoring of miRNome may help in several steps of clinical management, as reported in the representative studies cited below.

Early diagnosis and detection: miRNAs represent promising non-invasive biomarkers for the diagnosis of CRC. They can be detected in blood or other body fluids, enabling early diagnosis of the disease. Some miRNAs, such as miR-21, miR-92a, and miR-135b, were found to be upregulated in CRC patients [112,113,114,115].Prognosis and risk stratification: aberrant expression of specific miRNAs was associated with the prognosis and survival of CRC patients. Among them, miR-21 and miR-29a were correlated with a poor prognosis and shorter overall survival in CRC [116,117]. This information may be useful for identifying any patients who may need more aggressive treatment or closer monitoring.Prediction of treatment response: miRNAs can help to predict a patient’s response to a specific treatment. For example, some miRNAs, such as miR-21 or miR-17, have been related to resistance or sensitivity to chemotherapeutic agents commonly used in the treatment of CRC, such as 5-fluorouracil (5-FU) or oxaliplatin [118,119].Monitoring of treatment response and relapse: changes in miRNA expression levels during treatment can be monitored to evaluate treatment response and detect potential relapse. The monitoring of specific miRNAs, e.g., miR-21 or miR-155, can provide data on treatment efficacy and help in the process of personalizing approaches [118].

Importantly, although miRNAs hold promise for clinical applications in CRC, further research is required to validate their utility and establish standardized protocols for their use in clinical practice. By searching for keywords such as “COLORECTAL CANCER” and “miRNAs” on clinicaltrials.gov, we found eight clinical trials (shown in Table 1) that attempted to confirm efficacy, provide appropriate cut-off values, and determine the optimal methodologies for miRNA detection and analysis in CRC.

### 3.3. Lynch Syndrome and miRNAs

As already reported, LS can be explained by the mutation and/or epigenetic inactivation of MMR proteins causing MSI [120]. Several studies have shown how the deregulation of miRNAs is implicated in the pathogenesis of LS by interfering with the MMR system [121] and how this may lead to the identification of MSI status-specific signatures [122].

Balaguer et al. proposed the use of miRNAs as biomarkers, enabling the discrimination between sporadic MSI and LS via a microarray-based analysis conducted on a cohort of 74 patients. Among deregulated miRNAs, miR-30a, miR-16-2, and miR-362-5p emerged as the most significantly upregulated, while miR-1238 and miR-622 were found to be the most downregulated in LS with respect to sporadic MSI [123]. The clinical and biological importance of these miRNA classes is still under investigation.

The three main miRNAs that are upregulated in the LS and play a relevant role in dMMR, resulting in the development of MSI, are described below.

#### 3.3.1. miR-21

miR-21 was one of the first investigated miRNAs in mammals [124], and it is characterized as an oncogenic miRNA involved in the onset and progression of different malignancies [125,126,127]. How a high expression of miR-21 induces invasion and metastasis in CRC cells as a negative regulator of the tumor suppressor Programmed Cell Death 4 (PDCD4) [128] and phosphatase and tensin homolog (PTEN) has been described [129]. In addition, miR-21 has been shown to directly target the 3′UTR region of the MSH2 WT and MSH6 WT mRNAs, downregulating protein expression and ultimately causing a defect in damage-induced G2/M arrest and apoptosis in CRC cell lines. The downregulation of MSH2 and MSH6 causes heterodimerization failure in the MMR core, promoting an increase in mutational rates, MSI, and tumor progression (Figure 3A) [130].

#### 3.3.2. miR-137

miR-137 is one of the most important miRNAs as it regulates multiple aspects of neurogenesis [131]. It has been shown in many studies how hypermethylation of the gene encoding for miR-137 could cause its downregulation in CRC [132], suggesting a tumor suppressor role [133]. However, by correlating normal tissues and tissues from LS patients for differentially expressed microRNAs, miR-137 was found to be upregulated, suggesting that it might play a crucial role in the development of LS [134]. According to Liccardo et al., this result occurs due to the identification of the heterozygous substitution of a single base c.*226A>G in the 3′UTR region of the MSH2 mRNA. This mutation is able to induce the overexpression of the MSH2 protein to create a new binding site [135]. In fact, in this region, miR-137 has been shown to reduce the expression level of the MSH2 MMR core protein through the binding of the 3′UTR (Figure 3B) [136].

#### 3.3.3. miR-155

miR-155 is considered a multifunctional miRNA that is involved in numerous biological processes, and its overexpression has been reported in the development and progression of many human diseases [137,138]. miR-155 has relevant effects on the MMR system and, as a result, on MSI. This is because miR-155 acts by causing the downregulation of the MLH1, MSH2, and MSH6 proteins. The action of miR-155 combined with the loss of function of the MMR core proteins themselves, which are already unable to form heterodimers, results in a significant increase in mutation rates (Figure 3C). In addition, Valeri at al., through a retrospective study of CRC tissue samples with MLH1 gene mutation or MLH1 promoter methylation, demonstrated that miR-155 expression level showed a two- or- three-fold increase compared with healthy tissue. They hypothesized that MSI tumors with unknown dMMR may be dependent on the overexpression of miR-155 [139]. These findings assume considerable relevance in the prospective use of the evaluation of miR-155 expression as a complementary diagnostic test to be proposed when the current standard-of-care procedures fail in providing a clear diagnosis.

#### 3.3.4. Other miRNAs Player in Lynch Syndrome

In addition to the three well characterized miRNAs mentioned above, other player have been shown to be hypermethylated in LS-related CRC tissues compared with normal tissues: miR-132 and miR-345 are associated with dMMR in CRC, while miR-129-2 is associated with sporadic CRC with MMR deficiency by LS-related CRC [140]. miR-132-3p is downregulated in CRC cells [141] and is associated with a poor prognosis in CRC patients [142]. The upregulation of miR132-3p inhibits the proliferation, migration, and invasion of CRC cells [141,143]. miR-345 is significantly downregulated in the tissues of CRC patients, and this appears to cause a higher rate lymph node metastasis. These data were reversed by the upregulation of miRNAs [144]. Hypermethylation of the miR-192-2 promoter is reportedly linked to the onset of lymph node and liver metastasis in patients with CRC [145].

### 3.4. miRNAs Correlated with CRC and MSI Status

The relevance of MSI status and miRNA profiling was studied by Lanza et al. [30]. In their study, 39 CRC samples were analyzed via microarray analysis to find out the differences in miRNA expression between two groups of samples: 23 CRC MSS and 16 CRC MSI-H [30]. Based on this study, Earle et al. sought to associate the expression profile of miRNAs with MSI status [146]. The analysis included 55 formalin-fixed paraffin-embedded (FFPE) samples, 8 MSI-L, 25 MSS CRC, and 22 MSI-H adenocarcinomas, 6 of which were MSI-H CRC patients clinically diagnosed with LS. The miRNA assay showed 22 miRNAs, 20 of which were differentially expressed (11 overexpressed in CRC and 9 underexpressed in CRC). Specifically, MSI-L was associated with miR-92, let-7a, and miR-145, while MSI-H was associated with an increased relative expression of miR-155, miR-223, miR-31, and miR-26b. Only one miRNA was associated with MSS (miR-196a) [146].

The miRNAs that were associated with MSI-L, MSI-H, and MSS are described below.

#### 3.4.1. miRNAs and MSI-L Status

##### miR-92

miR-92 is associated with MSI-L status [146]. The upregulation of miR-92a resulted in 78.00% of cases in a cohort of 82 patients compared with adjacent normal tissues, correlating a with poor prognosis [147]. miR-92 is significantly elevated in the plasma of patients with CRC, demonstrating its potential as a noninvasive molecular marker for CRC screening [148,149].

##### let-7

let-7 is downregulated in CRC with MSI-L status [146,150]. As shown, many members of the let-7 family play a role in CRC. For example, let-7c suppresses tumor growth and metastasis [151], while an increased expression of let-7e-5p in CRC cell lines leads to reduced cell migration and proliferation [152] and increased sensitivity to 5-FU treatment [153]. A potential biological mechanism has been proposed through which let-7b binds the 3′UTR of the TGFBR1 messenger, affecting its expression with consequences that promote tumor growth in patients with MSS LS [31].

##### miR-145

miR-145 suppresses cell migration and invasion [154] and plays a negative regulatory role in colon cancer proliferation [154]. A reduced expression of miR-145-5p has been demonstrated in CRC tissue compared with normal tissue [155] and has been associated with invasion and premalignancy [154]. In addition, decreased miR-145 expression is reportedly associated with shorter survival time and increased disease recurrence [155]. In contrast, miR-145 has been found to be upregulated in metastatic CRC cell line models and has been associated with increased proliferative potential [155].

#### 3.4.2. miRNAs and MSI-H Status

##### miR-26b

miR-26b has been correlated with MSI-H status and promotes invasiveness, migration, and a stem cell-like phenotype in CRC cells [156]. In addition, miR-26b is significantly overexpressed in CRC patients [157] and is associated with lower overall survival [158]. The expression levels of miR-26b have also been associated with the mechanism of resistance to cetuximab [159].

##### miR-31

miR-31 is found overexpressed in CRC and in tissue samples from patients with CRC [160]. It has been shown that miR-31 promotes cell proliferation, invasion, and migration in vitro and tumorigenesis and metastasis in CRC [161], and it was associated with poor overall survival and progression-free overall survival in a meta-analysis of 4720 patients [162]. In addition, miR-31 is overexpressed in CRC among patients with LS, although no data are available on its role in pathology [146]. miR-31 has been demonstrated as a potential noninvasive biomarker of lymph node metastasis in patients with CRC [163]. However, in another study, serum miR-31 was reported to be downregulated in most patients with stage II and III CRC [164].

##### miR-223

miR-223 is upregulated in CRC tissues compared with adjacent normal mucosa [165] and is associated with a low survival rate in patients compared with those with low miRNA expression [166]. In addition, miR-223 is overexpressed in patients with LS [146]. miR-223 promotes doxorubicin resistance in CRC cells [167]. The downregulation of miR-223 inhibits the proliferation, migration, and invasion of CRC cells [165]. These results suggest that miR-223 could be a potential therapeutic target in patients with CRC [166]. In addition to these data, the quantification of miR-223 levels in plasma could be an accurate and reliable biomarker for the early identification and prognosis of CRC [168,169].

#### 3.4.3. miRNAs and MSS Status

##### miR-196a

The aberrant expression of miR-196 correlates with a poor prognosis in patients with CRC [170] and has been shown to correlate with MSS status [146]. miR-196 could be a promising biomarker for early diagnosis and prognosis in patients with CRC. Indeed, circulating miR-196b in the serum sample of patients with CRC has been shown to be significantly higher than in healthy controls, with a specificity of 63% and sensitivity of 87.38% [171]. The overexpression of miR-196b has also been found to promote resistance to chemotherapeutics such as 5-FU [170].

## 4. Current Clinical Scenario and Future Perspectives

Once the diagnosis of LS is reached via the germline testing of the genes involved in the MRR system, it is necessary to perform genetic tests on all family members at risk [14]. Patients with LS have a 50% probability of carrying the mutation to their children. In addition, considering the higher risk for LS patients of developing malignancies with respect to the general population, they should undergo specific and more frequent screening checks [14]. However, until now, the recommendations of screening guidelines have not been uniform [16].

While the most frequent malignancy arising in patients with LS is CRC, prospective studies with long-term follow-up have shown that colonoscopy evaluations can significantly reduce the incidence and mortality associated with CRC [172]. These data suggest that colonoscopy screening, if performed at regular and close intervals, can be considered as a key early intervention in LS [65,173]. The best age to start colonoscopy screening has not yet been decided [174]. The guidelines of The American Society of Clinical Oncology (ASCO), The European Society for Medical Oncology (ESMO), the National Comprehensive Cancer Network (NCCN), the U.S. Multi-Society Task Force on Colorectal Cancer (MSTF), The American College of Gastroenterology (ACG) [4,175], and others agree that screening should start at ages between 20 and 25 years or 2–5 years earlier than the youngest person in the family to be diagnosed with CRC, if diagnosed before age 25 [14,65].

In contrast, the optimal interval between one colonoscopy and the next should be 1–2 years [176,177]. The reason for this is because it has been observed that, in patients with LS, neoplastic lesions are identified within two years of the last negative examination because of the “immediate invasive growth capacity” of these tumors [82].

As mentioned above, several hypotheses have been explored to explain this phenomenon:Adenomatous polyp may initially develop with classic biallelic loss of APC and subsequently cause dMMR, leading to the accumulation of mutations fundamental to tumor growth [53].Adenomatous polyps may lose MMR early and subsequently acquire frameshift mutations in the single nucleotide repeats of the RNF43 gene, inducing malignant transformation [178].The traditional adenomatous polyp stage could be completely bypassed [55].

It should be noted that, even in screening, the gene-specific approach is becoming increasingly diffuse [16]. For example, in patients with MSH6 and PMS2 mutations, compared with patients with MLH1 and MSH2, the start of screening colonoscopies could be delayed for approximately 10 years because of the lower risk of developing CRC [176].

In addition to the high risk of developing CRC, LS patients with germline mutations of the MLH1, MSH2, and MSH6 proteins also have an increased risk of developing extra-colonic cancers such as endometrial, ovarian, gastric, pancreatic, and small bowel cancers [12].

For women with LS, the most common malignancies after CRC are endometrial and ovarian cancer [65]. Although data in this regard are controversial on specificity, sensitivity, reduced incidence, and mortality [65], in women who have had LS from the age of 30, guidelines recommend a transvaginal ultrasound with endometrial biopsy and hematic CA 125 assay for annual screening [175,177,179].

Unfortunately, to date, there is a lack of consistent data on the efficacy of screening for other LS-associated cancers, including gastric, pancreatic, and small bowel cancers [12].

For example, the ACG guidelines recommend performing esophagogastroduodenoscopy with a biopsy and concurrent testing for *Helicobacter pylori* starting at age 30–35 years, and this is repeatable every 5 years if there have already been cases of GC in the family of LS patient [4,173].

In cases of pancreatic cancer, despite the high risk, there are conflicting viewpoints [12,180]. According to an international group of experts, annual screening via endoscopic imaging modality (MRI), and/or ultrasound screening is recommended for patients with LS and at least one first-degree family member with pancreatic cancer [181]. However, according to NCCN or GCA guidelines, independent screening for pancreatic carcinoma is currently discouraged [12].

In patients with LS, because the risk of developing small bowel cancer is relatively low, there are no recommendations for using video capsule endoscopy screening tests [12,182].

### 4.1. Overview of Treatment in the Era of Precision Medicine

LS-associated tumors have a more favorable prognosis and are closely correlated with tumor stage compared with their sporadic counterparts [183]. MSI, in addition to being used as a positive prognostic biomarker, is emerging as a predictive biomarker of response to immune checkpoint inhibitors as well as a biomarker of resistance to classic cytotoxic chemotherapy regimens [77,184,185].

Resistance to classical chemotherapy regimens is caused by the absence of enzymes capable of recognizing the damage caused to the tumor cell by the drug in dMMR/MSI tumors [186]. This finding radically changed the management of patients with LS-related cancers [187]. For example, in the early stages of LS-associated CRC, the surgical strategy remains the first approach, mainly by subtotal or total colectomy because of the risk of developing metachronous neoplasms [63,188].

On the other hand, in the adjuvant setting, there are no recommendations for any chemotherapy treatments in patients with stage II CRC and MSI^+^, regardless of the presence or absence of unfavorable prognostic risk factors [173]. Data have shown that adjuvant 5-FU-based therapy for individuals with stage II CRC undergoing radical surgery has no impact on survival [189,190]. In contrast, in a larger study, subjects with stage II CRC treated with 5-FU appeared to have a worse prognosis [191].

This is different in the case of surgically treated stage III CRC with dMMR/MSI^+^ lymph nodes [192]. Like sporadic CRC, adjuvant polychemotherapy treatment is required, according to the FOLFOX (5-FU, folinic acid, or leucovorin and oxaliplatin) or CAPOX (capecitabine and oxaliplatin) [193]. This therapy has a significant impact on both overall survival and recurrence rates [194]. For patients who are not candidates for oxaliplatin therapy because they suffer from, for example, peripheral neuropathy, there may be no benefit from treatment with fluoropyrirnidine alone (5-FU or capecitabine) [82].

The revolutionary advancements in the treatment of patients with LS-related tumors has occurred alongside the development of immune checkpoint inhibitors. These act by manipulating and over-regulating the immune system, exploiting the biology underlying LS and other MSI-H/MMR tumors [65].

### 4.2. Immunotherapy

The relationship between cancer and the immune system (IS) has always been well known, but our understanding of the mechanisms is still developing [46,195]. Nowadays, it is known that the IS plays a dual role: on the one hand, it protects, and on the other hand, it promotes the process of carcinogenesis; in fact, among the hallmarks of cancer, we find both immune escape and tumor inflammation due to lymphocytic infiltrate [196].

LS-related tumors, particularly CRC, have high immunogenicity due to numerous neoantigens being recognized as extraneous and capable of triggering a highly strong immune response [65]. The dMMR leads to the accumulation of insertions and/or deletions in the microsatellite region and loci of the coding regions for tumor suppressor genes, thus initiating the process of carcinogenesis [197]. But the typical mutant phenotype of LS is also characterized by the accumulation of mutations in other coding regions of DNA, leading to the synthesis of novel frameshift peptides (FSPs) which act as tumor-specific antigens, triggering the strong inflammatory response of T lymphocytes [58,198]. Interestingly, FSP-reactive peripheral T lymphocytes have been identified in LS patients without CRC but not in non-LS individuals or CRC patients with dMMR tumors. This demonstrates that patients with LS are autoimmunized against FSP neoantigens before cancer formation, and this may explain the improved survival and reduced rate of distant localization observed in LS-CRC compared with sporadic CRC [199,200].

The process by which these tumors escape the influence of the IS is known as immunoediting, and the process consists of 3 main steps: elimination, equilibration, and escape [201]. In the specific case of LS-related CRC carcinogenesis, in the beginning, the IS is able to keep malignant cell foci under control (elimination phase), but the increasing accumulation of mutations leads to defense mechanisms to avoid immune destruction (equilibrium phase) and inevitable progression to cancer (escape phases) [46].

During the phase of elimination, cytotoxic T lymphocytes (CD8+ T cells) recognize FSP neoantigens through the major histocompatibility complex (MHC-I) class I on the surface of tumor cells, which in turn can trigger an immune response with an increased density of tumor-infiltrating lymphocytes [202].

During the equilibrium phase, tumor cells upregulate their immune checkpoint proteins, including programmed death 1 (PD-1) and programmed death ligand 1 (PD-L1) [203,204].

During the escape phase, the binding of PD-1 to activated T cells by PD-L1 induces the T cell into a quiescent state through the downregulation of signaling by proximal T cell receptor kinases and altered T1 cell metabolism, so tumor cells escape immune system-mediated elimination and unlimited inflammation is prevented [14].

In the treatment of LS-related cancers, especially CRC and cancers with MSI-H, anti PD-1 immune checkpoint inhibitors act on quiescent T cells and target proteins that overexpress the immune checkpoint, favoring the elimination of tumor cells [205].

Today, the treatment of CRC with MSI in the metastatic setting is based on the use of Pembrolizumab (anti PD-1) in the first-line setting (Figure 4A) [206]. In the second-line setting, on the other hand, the use of Nivolumab (anti PD-1) plus Ipilimumab (monoclonal antibody targeted against CTLA-4) has recently been approved for patients already treated with a platinum-, fluoropyrimidine-, and irinotecan-based regimen (Figure 4B) [207]. As for the neoadjuvant and adjuvant setting (such as NCT02912559), studies are still ongoing to evaluate the use of immune checkpoint inhibitors [208].

Based on the previously discussed data, it is therefore understood that there is a strong association between dMMR and response to immune checkpoint inhibitors, regardless of the site and tissue involved in the process of carcinogenesis [184]. These findings led to the recognition of TMB and immunoscore (tumor lymphocyte infiltrate) as new biomarkers for LS and LS-related cancers [209,210].

The immune escape mechanism, overexpression of immune checkpoint proteins, and quiescence of T lymphocytes and FSP neoantigens have provided a starting point for the development of primary prevention mechanisms for LS-associated cancers, as evidenced by vaccines that have already completed phase I/IIa [211,212].

Several strategies have been explored for vaccine development, including the ex vivo loading of dendritic cell peptides derived from patient monocytes [213]. The most compelling strategy is the delivery of cytotoxic T lymphocytes engineered to express a chimeric antigen receptor that recognizes a special antigen (CAR-T cells), creating engineered T cells with an engineered T cell receptor (TCR) or CAR against neo-FSP antigens [46,184].

The pivotal role of miRNAs in cancer cell immune escape, the modulation of the tumor microenvironment, and response to immunotherapeutic drugs has been established [214,215]. However, it is not yet clear whether they participate as immunomodulators or regulators of immunotherapy response in LS. Therefore, contemporary research should aim to shed light on this aspect.

### 4.3. Potential Use of miRNAs as Biomarkers or Therapeutic Targets

The study of miRNomes has provided novel insights, highlighting the importance of miRNAs as strategic therapeutic targets. Their small size and endogenous origin, which serve to reduce toxicity, would make them perfect candidates for the development of new anti-cancer therapies [216]. In addition, their ability to target multiple genes makes them a better treatment option than gene therapy [217]. Thus, taking advantage of their dual role in carcinogenesis, their utilization could involve two approaches: the first would involve the antisense nucleotide-mediated inhibition of oncogenic miRNAs, and the second would involve using mimetic miRNAs or miRNAs encoded by viral vectors that are intended to overcome the downmodulation of tumor suppressor miRNAs [218].

Different experimental findings have demonstrated the relevance of studying and monitoring miRNAs in certain types of cancer [219]. The discovery of the presence of circulating nucleic acids (CNAs) and their high stability in blood has opened the possibility of considering miRNAs as noninvasive biomarkers of diagnosis or drug efficiency [220,221]. Circulating miRNAs (c-miRs) have been shown to have potential in monitoring during clinical surveillance or the management of LS patients with dMMR. In addition, systemic c-miRNome analysis enables one to distinguish cancer-free path_MMR carriers from non-LS controls [222].

Strong evidence suggests that anti-miR-21 therapy might be promising in cases of LS and, as a result, for CRC [223]. High levels of miR-21 have been shown to correlate with poor diagnoses in CRC patients because it promotes the invasiveness of APC-mutated tumors [224]. Valeri et al. demonstrated how high levels of miR-21 are associated with an acquired mechanism of resistance to 5-FU treatment [130]. In addition, a correlation between miR-21 levels in both tumor and serum was demonstrated, revealing how miR-21 levels decreased after the surgical removal of CRCs [225]. The potential use of miR-21 as a powerful non-invasive biomarker for the early detection of CRC neoplasm development and treatment response has been further corroborated [225,226,227]. However, due to the limited number of studies, further efforts are required to translate the use of wider panels of miRNAs as biomarkers for the clinical management of LS. In this regard, the recent introduction of next-generation non-invasive technologies, including PCR-free miRNA detection systems for point-of-care testing (POCT), which allow for an easier and more accurate quantification of c-miRs, could support a deeper investigation of the LS miRNome landscape, similarly to what was performed in a large cohort of patients from different cancer types [228].

In addition, the ability to differentiate miRNA sequences is an important factor in selecting candidate miRNAs with therapeutic potential [229]. In fact, the bias that needs to be overcome in order to improving treatment response is the close correlation in sequence regions that many miRNAs belonging to different families display [230]. Despite the scientific community’s efforts in producing increasingly promising data and results to promote research on the importance of the biological study and therapeutic potential of miRNAs [231], miRNA-based therapeutic approaches are still in their infancy. We recently reported that miR-221, an oncogenic miRNA with several implications in cancer onset and progression [24], can be successfully targeted in cancer patients by a miRNA inhibitor, LNA-i-miR221, which may have therapeutic activity in CRC in a First-in-human clinical trial [231].

In light of this, the possible development of an anti-miR-155 therapy could also become a promising therapeutic target since it has been demonstrated that miR-155 is able to promote proliferation and inhibit apoptosis in LS [232].

## 5. Conclusions

As described in this review, the molecular mechanisms behind the onset of LS are still under investigation, and new models of progression may be proposed along with deeper investigations into the genetic changes in the protein-coding and non-coding genes underlying the disease. This information will also lead to novel therapeutic decisions. At present, although most of the miRNome has been sequenced, the functional contributions of many miRNAs in the onset of human cancers are yet to be clarified, especially for LS-related tumors. Despite the fact that the role of miRNAs as biomarkers or potential targets for LS has been proposed, the majority of laboratory-derived evidence refers to CRC, while only a few older studies focus on LS. One of the aims of this review is to make the scientific community more conscious of the unmet needs for the implementation and use of miRNAs-based clinical approaches for this complex disease. The strong differences in the expression levels of miRNAs in LS patients compared to healthy individuals and their potential prognostic relevance suggests the necessity to move toward a deeper characterization of the already described miRNAs and the discovery of new key miRNA players in order to promote their use for clinical practice. This means that studies on MSI tumors with unknown dMMR or on LS patients who are unresponsive to standard therapeutic regiments but have a well-defined miRNomic profile will assume more relevance. This innovative approach assumes considerable importance in the era of precision medicine, especially for cancer patients that are refractory to the other currently available therapeutic regiments. Overall, despite the fact that, at present, information on the role of miRNAs in LS is still limited and no clinical trials focusing on the use of miRNAs are ongoing, miRNA represent an extremely powerful tool for generating novel biomarkers and for the design of novel treatments. Therefore, further efforts should be directed toward the discovery of new vulnerabilities in the miRNAome landscape to provide valuable information for the management of LS.

## Figures and Tables

**Figure 1 cancers-15-03930-f001:**
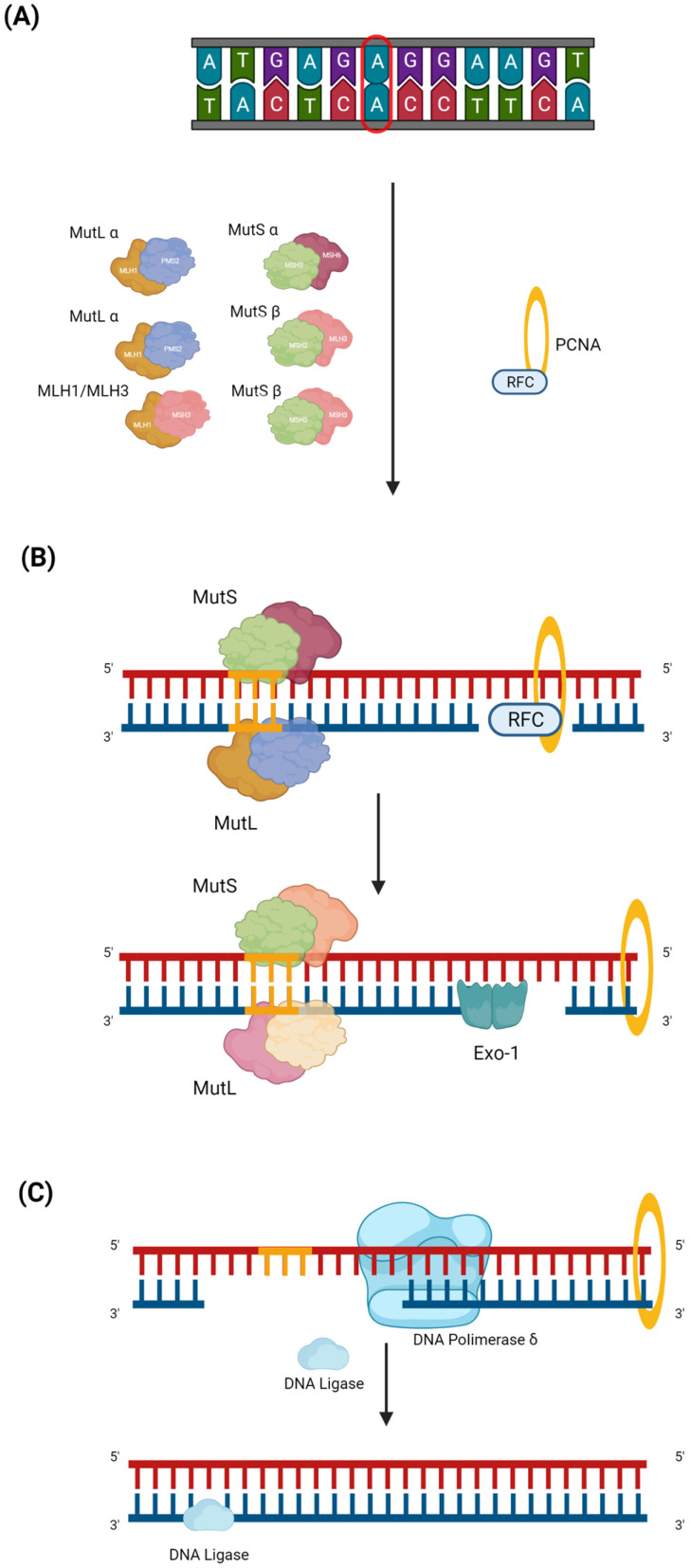
MMR pathway. Mismatch repair (MMR) is a DNA repair mechanism that is activated during the replication phase. Despite the high replication fidelity of polymerases, base insertion errors can be made, creating single nucleotide mismatches, insertions, and deletion loops in repeated stretches of DNA, e.g., in the microsatellite region. The MMR can be divided into three steps. (**A**) The first is the damage recognition step by the MutS complex, which consists of MutS α (MSH2 and MSH6) or MutS β (MSH2 and MSH3). After recognizing the damaged region, the MutL (MutL α (MLH1 and PMS2), MutL β (MLH1 and PMS1), or MutL γ (MLH1 and MLH3)) complex is formed. The MutS and MutL heterodimeric complexes bind to the damaged strand region, and recruiting PCNA is transported by RFC, which, once it has bound the DNA, is ejected. (**B**) During the second step, EXO-1, an enzyme belonging to exonucleases which acts to mediate base excision, is recruited. (**C**) In the last step, first, DNA polymerase *δ*, which latches onto the PCNA closed around the DNA, intervenes to properly synthesize the mismatch; then, DNA ligase 1 resynthesizes the bonds between the bases.

**Figure 2 cancers-15-03930-f002:**
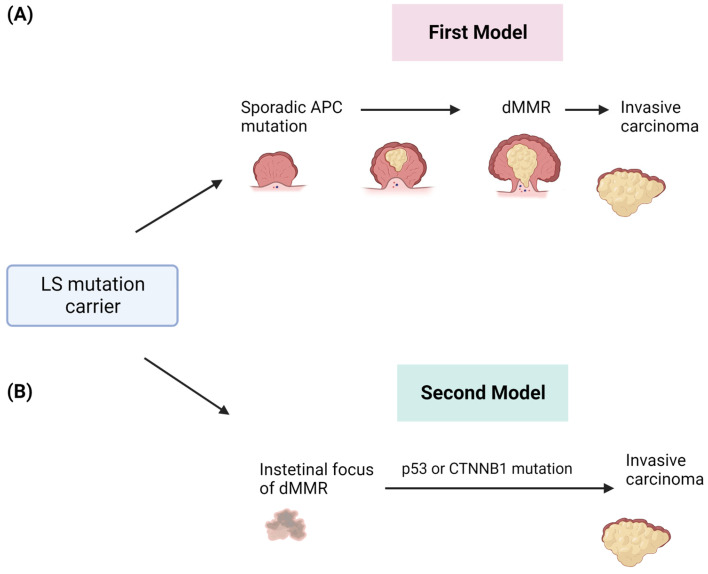
Two different models of CRC development in patients affected by LS. (**A**) The first model involves the sporadic acquisition of early APC gene mutation in adenomas and after the onset of dMMR and invasive tumors. (**B**) The second model describes the early onset of dMMR and the direct development of invasive tumors.

**Figure 3 cancers-15-03930-f003:**
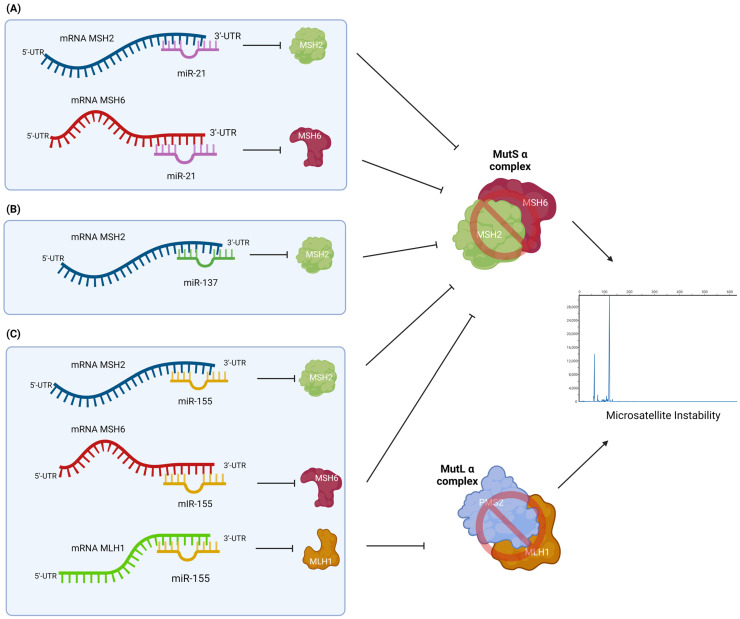
Mechanism of action of miRNAs in Lynch syndrome. LS can be explained by mismatch repair deficiency (dMMR), but several studies have shown how other factors, such as different expressions of certain miRNAs, can interact, causing the deregulation of mRNAs encoding for MMR proteins. (**A**) miR-21 has a dual target. One is MSH2, and the other is MSH6. In both cases, if upregulated, miR-21 interacts with the 3′ UTR of both mRNAs, omitting the formation of the MutS α complex and promoting tumor progression. (**B**) miR-137 was also found to be upregulated in LS. The upregulation of miR-137 causes a decrease in the expression of the MSH2 protein due to its ability to bind the 3′UTR of mRNA. The absence of the MSH2 protein results in the failure to assemble the MutS α or MutS β complexes. (**C**) miR-155 has several target mRNAs to which it binds in the 3′UTR region. In fact, when miR-155 is overexpressed, it deregulates both the expression of the proteins that form the MutS α complex (MSH2 or MSH6) and the MLH1 protein, which instead forms the MutL α complex together with the PMS2 protein. In each case, the failure to form complexes causes an absence of MMR and the epiphenomenon of microsatellite instability (MSI).

**Figure 4 cancers-15-03930-f004:**
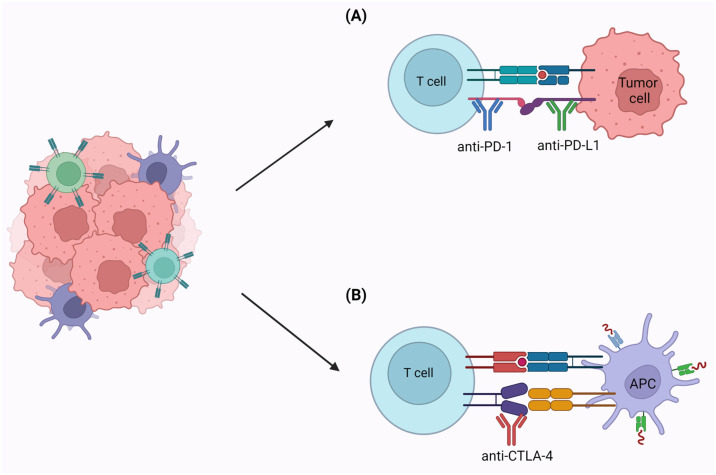
Immunotherapy. LS-related tumors have high immunogenicity due to their ability to generate neoantigens targeted by the immune system (IS), which recognizes them as non-self antigens and destroys the tumor cells. The tumor is able to mask the neoantigens from the IS through an immune escape mechanism defined as immunoediting. The use of immunotherapy in clinical practice to bypass immunoediting promotes the elimination of tumor cells through two approaches: (**A**) Direct, through checkpoint inhibitors such as Pembrolizumab and Nivolumab (ant PD-1), which act on quiescent T cells and overexpressed target proteins, and (**B**) Indirect, with Ipilimumab (monoclonal antibody targeting CTLA-4), which externally blocks CTLA-4 mediated inhibition signals on cytotoxic T lymphocytes.

**Table 1 cancers-15-03930-t001:** Clinical trials available on clinicaltrial.gov (listed in order of Last Update Posted from most recent to oldest).

Study Title	ID	Status	Last Update Posted
Project CADENCE (CAncer Detected Early caN be CurEd) (CADENCE)	NCT05633342	Recruiting	20-03-2023
Predictive and Prognostic Value of Inflammatory Markers and microRNA in Stage IV Colorectal Cancer	NCT04149613	Enrolling by invitation	20-07-2022
Establishment of Molecular Classification Models for Early Diagnosis of Digestive System Cancers	NCT05431621	Recruiting	24-06-2022
Validation of a microRNA-based Fecal (miRFec) Test for Colorectal Cancer Screening	NCT05346757	Recruiting	26-04-2022
Timisnar-Biomarkers Substudy (Timisnar-mirna)	NCT03962088	Recruiting	28-07-2021
microRNAs Tool for Stratifying Stage II Colon Cancer	NCT0263508	Recruiting	30-12-2015
A 6 microRNA Tool for Stratifying Stage II Colon Cancer of Receiving Adjuvant Chemotherapy	NCT02466113	Not yet recruiting	09-06-2015
Quantifying Micro RNA Levels of Colon (CRC)	NCT01712958	Unknown status	24-10-2012

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
