# Peer review of "Lynch Syndrome Biopathology and Treatment: The Potential Role of microRNAs in Clinical Practice"

_cancers, 2023, doi:10.3390/cancers15153930_

Round 1
Reviewer 1 Report
<Overview>
In this concise review, the authors provided a brief overview of Lynch syndrome (LS) and its clinical management, ranging from its current classification, diagnosis to potential treatment strategies. They also presented a few examples of past efforts to identify microRNAs that promote LS pathogenesis, particularly those that regulate the DNA mismatch repair (MMR) pathway.
Although the authors have thoughtfully laid down several aims in the abstract of this review, the key implications and interconnectivity of these aims have not been fully revealed and dissected to make this work a thought-provoking review for a wider readership.
Major comments:
1. The various sections of this review are very well structured and it is evident that the authors made an effort to pinpoint the limitations of current classification/diagnosis of various LS-associated malignancies/cancers and sporadic CRCs. In my opinion, in-depth discussions on how detecting or targeting miRNAs can complement or enhance current standard-of-care in the diagnosis and treatment strategies of these cancers will help the readers to better understand the clinical potential of miRNAs. For instance, would the miRNA biomarkers complement or outperform immunohistochemistry testing and MSI testing for LS?
2. The miRNA-related studies cited in this manuscript (section 4.3) are mainly based on published reviews. How much of the primary data was derived from lab-based experiments and are these findings readily translatable to clinical applications? Also, Reference 180 (a fairly dated JNCI article) reported findings from a relatively small case-control cohort. Do the authors truly believe that the study was sufficiently powered to derive robust miRNA signatures for CRC diagnosis and prognosis? The authors could map these studies against the best practices of miRNA-based diagnostics development (such as those described in PMID: 34999430) to discuss whether the key considerations of discovery and development of LS-associated miRNA biomarkers have been fulfilled (or can we do more to develop better LS-associated miRNA biomarkers).
Minor comments:
1. The use of visuals like schematic diagrams/ tables/ flow charts could help to explain complex processes involved in driving LS onset/progression, making diagnosis and formulating treatment strategies for LS more effectively.
2. Line 118/119: Clumsy sentence structure. The authors are strongly encouraged to rephrase this sentence.
3. Line 131: “fast arise” could be replaced by” accelerated development of”.
4. Section 3.1/lines 255-258: are the miRNA coding sequences hotspot sites of dMMR? If yes, the authors can seize the opportunity to discuss its implication(s).
5. Line 474: “SI” is a typo error and should be replaced by “IS (immune system)”?
6. Lines 482-484: Can the authors state which cell type(s) harbour the upregulation of immune checkpoint proteins and clarify whether miRNAs are involved in driving such processes (plus how can we intervene)?
7. Lines 549-550: Clumsy sentence structure. The authors are strongly encouraged to rephrase this sentence.
The authors will only need to rephrase a few sentences.
Author Response
Reviewer #1: In this concise review, the authors provided a brief overview of Lynch syndrome (LS) and its clinical management, ranging from its current classification, diagnosis to potential treatment strategies. They also presented a few examples of past efforts to identify microRNAs that promote LS pathogenesis, particularly those that regulate the DNA mismatch repair (MMR) pathway.
Although the authors have thoughtfully laid down several aims in the abstract of this review, the key implications and interconnectivity of these aims have not been fully revealed and dissected to make this work a thought-provoking review for a wider readership.
Major comments:
- The various sections of this review are very well structured, and it is evident that the authors made an effort to pinpoint the limitations of current classification/diagnosis of various LS-associated malignancies/cancers and sporadic CRCs. In my opinion, in-depth discussions on how detecting or targeting miRNAs can complement or enhance current standard-of-care in the diagnosis and treatment strategies of these cancers will help the readers to better understand the clinical potential of miRNAs. For instance, would the miRNA biomarkers complement or outperform immunohistochemistry testing and MSI testing for LS?
Response 1: We would like to thank the Reviewer for this important comment.
To date, the gold standard for the diagnosis of LS is represented by immune-histochemistry analysis enabling the detection of dMMR, and by the PCR-based analysis enabling the detection of MSI status in short repeated sequences at the 5 major loci. Unfortunately, to the best of our knowledge, there are no studies highlighting the use of miRNAs as biomarkers to be associated to these specific approaches in clinics for LS treatment (such. as ongoing clinical trial). In Section 4.3, we have mentioned several studies demonstrating the high potential of miRNAs in supporting prognosis and diagnosis in many cancer diseases, including LS. For example, we highlighted the relevance of circulating nucleic acids (CNAs) in clinical practice (see miR-21 –page 10, section 3.3, lines 389-398), as potential non-invasive biomarker of diagnosis or in evaluating response to therapies. In the present version of the manuscript, we have tried to emphasize the role of miRNAs as biomarkers, and the association between miRNAs and evaluation of MSI status according to reviewer suggestion, describing most relevant findings:
Page 9, section 3.3, line 377-384
Page 10, section 3.3, line 422-425
- The miRNA-related studies cited in this manuscript (section 4.3) are mainly based on published reviews. How much of the primary data was derived from lab-based experiments and are these findings readily translatable to clinical applications?
Response 2: Unfortunately, only few preclinical (and nonclinical) studies specifically focusing on miRNAs in LS are available in literature, to the best of our knowledge. To address the point highlighted by the Reviewer, we critically highlighted the current information (page 16-17 section 4.3 line 702-720) and in Conclusion section (page 17 line 733-741).
Minor comments:
1. The use of visuals like schematic diagrams/ tables/ flow charts could help to explain complex processes involved in driving LS onset/progression, making diagnosis and formulating treatment strategies for LS more effectively.
Response 1: As suggested, a schematic diagram of LS progression has been added as new Figure 2.
2. Line 118/119: Clumsy sentence structure. The authors are strongly encouraged to rephrase this sentence.
Response 2: We rephrased as follow: “The malignant transformation from LS to CRC has been described as a multistep process, which has been explained by different model of progression (Figure 2). The first proposed model, supposes that tumors arise from polypoid lesions with a proficient MMR whose deficiency will occur later.”
3. Line 131: “fast arise” could be replaced by” accelerated development of”.
Response 3: We replaced it as suggested.
4. Section 3.1/lines 255-258: are the miRNA coding sequences hotspot sites of dMMR? If yes, the authors can seize the opportunity to discuss its implication(s).
Response 4: To our acknowledge, there are no clear-cut data to support this highly relevant implication to date.
5. Line 474: “SI” is a typo error and should be replaced by “IS (immune system)”?
Response 5: We replaced it as suggested.
6. Lines 482-484: Can the authors state which cell type(s) harbour the upregulation of immune checkpoint proteins and clarify whether miRNAs are involved in driving such processes (plus how can we intervene)?
Response 6: We specified that during the equilibrium phase, different immune checkpoint proteins are upregulated in tumor cells (page 15, section 4.2, line 634).
Moreover, specified in page 16, section 4.2, line 666-670: “The pivotal role of miRNAs in cancer cell immune escape, modulation of tumor microenvironment and response to immunotherapeutic drugs has been established [216, 217], however it is not yet clear whether they participate as immunomodulators or regulator of response to immunotherapy in LS. Therefore, the interest of current research should be to shed light on this field.”
7. Lines 549-550: Clumsy sentence structure. The authors are strongly encouraged to rephrase this sentence.
Response 7: We replaced as follow: “miRNA-based therapeutic approaches are still in their infancy.”
Reviewer 2 Report
The authors presented a Review on the molecular aspects linked to Lynch Syndrome with the aim of highlighting the role of microRNAs as potential disease biomarker and therapeutic targets.
Despite the Authors, presented several aspects related to Lynch Syndrome a very marginal part of the manuscript is devoted to describe the role of miRNAs in this disease. Specifically, only one page in the Chapter 3.2 is dedicated in illustrating the role of these molecules in Lynch Syndrome and many citations are not justified. In the current form the manuscript is not suitable for a publication in the Cancers journal.
Major points
- A very marginal description of the role of miRNAs in Lynch Syndrome or microsatellite instability is provided and many key publications are not described, including Kau et al., 2015 (doi: 10.1186/s13148-015-0059-3), Earle et al 2010 (doi: 10.2353/jmoldx.2010.090154), Xicola et al., 2016 (doi: 10.1093/carcin/bgw064), Lanza et al., 2007 (doi: 10.1186/1476-4598-6-54), or the most recent Sievänen et al., 2023 (doi: 10.1002/ijc.34338)
- Many references are not completely justified, including 22, 25, 26, 27, 28, 29, 36, 37. In the manuscript are present more citations related to miRNAs in multiple myeloma than those related to miRNAs in the Lynch Syndrome .
The quality of English language is adeguate.
Author Response
Reviewer #2: The authors presented a Review on the molecular aspects linked to Lynch Syndrome with the aim of highlighting the role of microRNAs as potential disease biomarker and therapeutic targets.
Despite the Authors, presented several aspects related to Lynch Syndrome a very marginal part of the manuscript is devoted to describe the role of miRNAs in this disease. Specifically, only one page in the Chapter 3.2 is dedicated in illustrating the role of these molecules in Lynch Syndrome and many citations are not justified. In the current form the manuscript is not suitable for a publication in the Cancers journal.
We want to thank the Reviewer for addressed this important point. We are aware about the low number of studies investigating the role of miRNAs in LS and the consequent limited information provided in this review. On the other hand, by considering the spread of miRNA-related information in preclinical and clinical studies regarding a huge number of tumors, including CRC, our aim was to underline the interest of scientific community toward a deeper investigation in the field, in the era of integromic studies and molecular medicine.
Major points
- A very marginal description of the role of miRNAs in Lynch Syndrome or microsatellite instability is provided and many key publications are not described, including Kau et al., 2015 (doi: 10.1186/s13148-015-0059-3), Earle et al 2010 (doi: 10.2353/jmoldx.2010.090154), Xicola et al., 2016 (doi: 10.1093/carcin/bgw064), Lanza et al., 2007 (doi: 10.1186/1476-4598-6-54), or the most recent Sievänen et al., 2023 (doi: 10.1002/ijc.34338).
Response 1: We want to thank the reviewer for highlighting these studies that helped us to provide a more comprehensive view of available data. We discussed each study in follow sections:
Page 2 Section 1. line 59-62
Page 10-11, Section 3.3 line 426-435
Page 11-12, Section 3.4 line 436-506
Page 16, section 4.3, line 688-691
- Many references are not completely justified, including 22, 25, 26, 27, 28, 29, 36, 37. In the manuscript are present more citations related to miRNAs in multiple myeloma than those related to miRNAs in the Lynch Syndrome.
Response 2: The mentioned references aimed to highlight the role of ncRNAs in cancer diagnosis and therapy (22, 27 [REF NEW 23, 24]). However, we thank the Reviewer for the suggestions, which aim to improve the relevancy of the manuscript. In this regard, we deleted the references regarding lncRNAs to focus on miRNAs (25, 26, 28, 29) and 36, 37, which refers to less recent studies.
Reviewer 3 Report
Lynch syndrome is characterized by an increased risk of developing colorectal cancer and other cancers. This review is presented clearly, with informative figures and up-to-date references. The authors have included sections to encompass various aspects of this condition comprehensively. I have a few comments:
1. page 4, line 163: replace extra-colic by extra-colonic
2. Please explain why it was selected only three miRNAs from so many others described as altered in colorectal cancer. Considering the title of the manuscript, miRNAs are little explored.
3. item 4: one paragraph about breast cancer in LS patients will be interesting.
The review is well-written, straightforward, and easy to follow.
Author Response
Reviewer #3: Lynch syndrome is characterized by an increased risk of developing colorectal cancer and other cancers. This review is presented clearly, with informative figures and up-to-date references. The authors have included sections to encompass various aspects of this condition comprehensively. I have a few comments:
1. page 4, line 163: replace extra-colic by extra-colonic
Response 1: We replaced it as suggested.
2. Please explain why it was selected only three miRNAs from so many others described as altered in colorectal cancer. Considering the title of the manuscript, miRNAs are little explored.
Response 2: We thanks the Reviewer for highlighting this important point We aimed to specifically focus on LS (be excluding for example sporadic CRCs and other forms). We are aware about the low number of studies elucidating the role of miRNAs in LS. Our aim was to pique the interest of scientific community toward a deeper investigation in the field. However, we agree that is important to provide additional information about the clearly established role of miRNAs in CRCs. In this regard we have added the following paragraph and the Table 1 placed in the text page 8-9, line 332-371.
We added novel information, just published (June 2023) on the potential antitumor activity of a first-in-class miR-221 inhibitor (LNA-i-miR-221) in a first-in-human clinical trial in cancer patients (page 17, section 4.3, line 710-720).
3. One paragraph about breast cancer in LS patients will be interesting.
Response 3: We acknowledge the relevance of this point. To discuss this issue, we added the paragraph as follow (page 6-7, section 2.3, line 258-281).
Reviewer 4 Report
The review by Ascrizzi et al. is a well-written paper which can be publishable in Cancers Journal.
Author Response
Reviewer #4: The review by Ascrizzi et al. is a well-written paper which can be publishable in Cancers Journal.
We thank the Reviewer for the positive evaluation of our manuscript.
Round 2
Reviewer 1 Report
Not applicable. The authors have now addressed my concerns.
Author Response
Reviewer #1: Not applicable. The authors have now addressed my concerns.
We thank the Reviewer for his relevant contribution to the improvements of our work.
.
Reviewer 2 Report
The authors address many of the points risen by the Reviewer. However, an excess of not justified self-citation is already present and must be removed. Specifically, citations from 127 to 129 concern the role of miR-21 in Multiple Myeloma which is out of the scope of the Review. A lot of reviews about this miRNA are available including the one of Jenike et al (doi: 10.1186/s40364-021-00272-1). Please indicate such Reviews and not articles whose context is far from the topic of the Review.
Author Response
Reviewer #2: The authors address many of the points risen by the Reviewer. However, an excess of not justified self-citation is already present and must be removed. Specifically, citations from 127 to 129 concern the role of miR-21 in Multiple Myeloma which is out of the scope of the Review. A lot of reviews about this miRNA are available including the one of Jenike et al (doi: 10.1186/s40364-021-00272-1). Please indicate such Reviews and not articles whose context is far from the topic of the Review.
We thank again this Reviewer for these additional suggestions. Accordingly, we have removed references 127 to 129 and added the recommended reference Jenike et al (doi: 10.1186/s40364-021-00272-1) [NEW REF 127].
Round 3
Reviewer 2 Report
The Authors addressed all the points risen during the revision process.